# Multi-Agent Robot System to Monitor and Enforce Physical Distancing Constraints in Large Areas to Combat COVID-19 and Future Pandemics

**Syed Hammad Hussain Shah \*** , **Ole-Martin Hagen Steinnes, Eirik Gribbestad Gustafsson and Ibrahim A. Hameed**

Department of Information and Communication Technologies (ICT) and Natural Sciences, Faculty of Information Technology and Electrical Engineering, Norwegian University of Science and Technology (NTNU), 6009 Ålesund, Norway; omsteinn@stud.ntnu.no (O.-M.H.S.); eirikgg@stud.ntnu.no (E.G.G.); ibib@ntnu.no (I.A.H.)
\* Correspondence: syed.h.h.shah@ntnu.no; Tel.: +47-4656-4323

**Abstract:** Random outbreaks of infectious diseases in the past have left a persistent impact on societies. Currently, COVID-19 is spreading worldwide and consequently risking human lives. In this regard, maintaining physical distance has turned into an essential precautionary measure to curb the spread of the virus. In this paper, we propose an autonomous monitoring system that is able to enforce physical distancing rules in large areas round the clock without human intervention. We present a novel system to automatically detect groups of individuals who do not comply with physical distancing constraints, i.e., maintaining a distance of 1 m, by tracking them within large areas to re-identify them in case of repetitive non-compliance and enforcing physical distancing. We used a distributed network of multiple CCTV cameras mounted to the walls of buildings for the detection, tracking and re-identification of non-compliant groups. Furthermore, we used multiple self-docking autonomous robots with collision-free navigation to enforce physical distancing constraints by sending alert messages to those persons who are not adhering to physical distancing constraints. We conducted 28 experiments that included 15 participants in different scenarios to evaluate and highlight the performance and significance of the present system. The presented system is capable of re-identifying repetitive violations of physical distancing constraints by a non-compliant group, with high accuracy in terms of detection, tracking and localization through a set of coordinated CCTV cameras. Autonomous robots in the present system are capable of attending to non-compliant groups in multiple regions of a large area and encouraging them to comply with the constraints.

**Keywords:** COVID-19; pandemics; physical distancing; disease prevention; contact tracking; enforcement; assistive robotics; autonomous systems; human–robot interaction (HRI)

## 1. Introduction

In the past few years, infectious diseases have been found to be very challenging and difficult to control due to their transferring effect, resulting in a large impact on society. They can spread at different geographical levels due to their ability of human-to-human transmission. According to the national public health institute named the Centers for Disease Control and Prevention (CDC) in the United States, an infectious disease can be declared as a 'pandemic' when a sudden and rapid increase in its cases is seen in the global population. In recent history, various pandemics have been reported. In a report created by Nicholas [1], the history of pandemics over a period of time has been explained with their impact in terms of death toll, which is summarized in Table 1.

**Table 1.** The history of pandemics with respect to the death toll.

| Disease Name | Time Period | Death Toll |
|---|---|---|
| Spanish Flu | 1918–1919 | 40 Million–50 Million |
| Hong Kong Flu | 1968–1970 | 1 Million |
| HIV/AIDS | 1981–Present | 25 Million–35 Million |
| SARS | 2002–2003 | 770 |
| Swine Flu | 2009–2010 | 200,000 |
| Ebola | 2014–2016 | 11,000 |
| COVID-19 | 2019–Present | 2.04 Million |

Based on the history of pandemics presented in this study [1], it is likely that more such pandemics will arise in the future. Therefore, nations should be prepared for them. It is imperative to understand the reasons behind the transmission of such infectious diseases. A large number of epidemiological studies have pointed out that the main path of the spread of such infectious diseases has been human-to-human transmission [2]. This indicates that infectious diseases spread when people maintain direct physical contact with each other. Physical distancing has always been recommended as the most effective safety measure to avoid the spread of such pandemics [3] and is currently being implemented by governments worldwide to slow the spread of the COVID-19 virus [4]. Despite the implementation of such measures, the span of the virus spread has been increasing with time due to the violation of set constraints because of lack of knowledge and carelessness. In this case, continuous monitoring is required to enforce the constraints to control the spread of the virus. It is difficult to manually monitor all areas; therefore, intelligent and autonomous systems are required for efficient and persistent monitoring of set constraints such as the use of facemasks, physical distancing, body temperature checks, etc. Moreover, modern interactive technology platforms such as robots hold potential to be used for the enforcement of those constraints through social interaction with people. Such approaches can be beneficial in reducing human-to-human interaction to potentially curb the spread of the virus. Robotics has a huge potential to play a vital role in the current fight against the COVID-19 virus [5]. Robots can be deployed for various purposes to help curb the spread of the virus. For instance, they can be utilized for mobile surveillance, disinfection, delivery, interactive awareness systems, companion robots, vital signs detection, etc. In past research, a wide range of multi-agent robot systems (MARS) based on heterogeneous distributed sensor networks have been proposed for effective and efficient surveillance in multiple scenarios [6,7]. MARS is a system that comprises fixed agents, i.e., sensors fixed at some location, and single or multiple mobile agents, i.e., robots.

During the current pandemic situation, performing operational tasks, such as surveillance, digital interaction, help desks and medical service provision, using robots has gained huge popularity [8]. Fan et al. [9] presented an autonomous quadruped robot to ensure physical distancing to combat COVID-19. The designed robot was supposed to roam around the place for persistent surveillance to detect violations of physical distancing constraints. In case of any violation, the robot informed the people through verbal cues to maintain a safe distance. Moreover, social robots are playing a vital role in combating COVID-19 by minimizing person-to-person interactions, especially in healthcare services [10–12]. Recently, Sathyamoorthy et al. [13] presented a robot system to monitor physical distancing constraints in crowds and enforce them through robots by displaying alert messages on the robot's mounted display. In the case of persistent non-compliance by a group of persons wandering from one place to another, the robot pursued that group and kept displaying the message. One of the limitations of this system is that while pursuing that group, there is a high possibility that the robot would not be able to attend to other non-compliant groups. Moreover, this study is missing the mechanism to track groups that remained unattended by the robot. Furthermore, there was no long-term tracking of non-compliant groups to further monitor their behavior after receiving the alert message from the robot. Consequently, it was not possible to track repetitive violations using this

system. Due to these issues, this system may not be able to effectively enforce physical distancing constraints in large areas such as shopping malls, airports, etc. In this regard, we present an autonomous and interactive monitoring system with large-scale area coverage for effective and efficient monitoring to combat COVID-19 and future pandemics.

In the present study, we present a cooperative MARS for monitoring and enforcing physical distancing constraints in large areas through human–robot interaction (HRI) to combat COVID-19 and future pandemics. In the present study, a group of persons who violate physical distancing constraints is referred to as a non-compliant group. The aims of the proposed system are as follows: (1) persistent monitoring of large indoor areas using multiple CCTV cameras to detect the violation of physical distancing constraints; (2) interactive encouragement of non-compliant groups to adhere to physical distancing constraints by giving them an alert message through speech-based HRI; (3) long-term tracking and re-identification of non-compliant groups through a multi-camera system to alert them on highest priority and report to the control room in the case of repetitive violations of physical distancing constraints. As shown in Figure 1, the design of the proposed system is based on two types of agents: (1) fixed agents, i.e., calibrated CCTV cameras, and (2) mobile agents, i.e., self-docking autonomous robots with collision-free navigation. Both agents work cooperatively by mutually sharing useful information between each other.

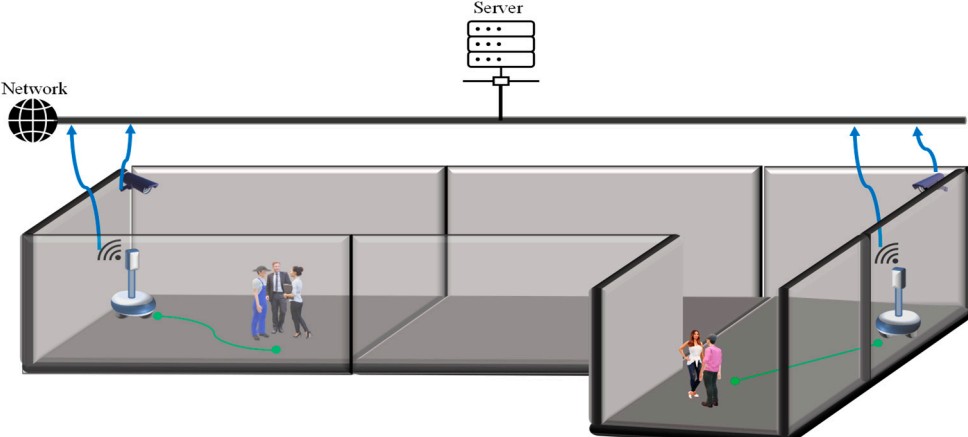

**Figure 1.** Multi-agent robot system.

In the present system, the persistent monitoring of physical distancing constraints is performed based on the visual information received from the distributed network of multiple CCTV cameras mounted within the building. The team of robots stay at their docking stations until a violation of physical distancing constraints is detected by the cameras. In the case of a detected violation, the system shares the location of the non-compliant group with the robot that is located closest to the area of the building where the violation was detected. After receiving the location, the robot navigates to the given location to convey an alert message to the target non-compliant group. The system architecture with a detailed description of each functional module is presented in Section 3. The main contributions of the present study are summarized as follows:

1. We propose an intelligent and cooperative MARS for the efficient monitoring of physical distancing constraints and interactively enforcing them through HRI to combat COVID-19 and future pandemics. To the best of our knowledge, we are the first to propose such a monitoring system, which is based on a distributed network of multiple cameras and a multi-robot system (MRS) to combat ongoing and future pandemics.
2. We develop a pipeline for group re-identification through person re-identification using a deep learning-based technique to track and re-identify non-compliant groups through the multi-camera system. This method ensures the long-term tracking of non-compliant groups that are wandering from one place to another in large areas,

attending to them at highest priority through a robot and notifying the security control room in a timely manner in case of repetitive violations.

3. Based on our proposed system, we ensured that all non-compliant groups were inclusively tracked and received the alert message about a breach of physical distancing constraints through HRI.

The rest of the paper is organized as follows. Section 2 provides an overview of the existing work related to multi-agent systems (MAS) with regard to surveillance, the effectiveness of physical distancing and the potential of robotics to combat COVID-19 and future pandemics. The proposed system with detailed descriptions of its modules is presented in Section 3. Evaluation metrics and experimental results are described in Section 4. Finally, we conclude our proposed system in Section 5 with a discussion about limitations and future directions.

## 2. Related Work

In this section, we review the previous literature on robotic systems specifically categorized into MAS for surveillance, the potential of robotics to combat COVID-19, the effectiveness of physical distancing and emerging technologies to monitor physical distancing.

### 2.1. Multi-Agent Systems for Intelligent Surveillance

Intelligent surveillance includes various tasks such as detection, tracking and understanding different behaviors in various environments [14]. MAS has gained huge attention in recent years due to its broad range of applications such as cooperative surveillance, distributed tracking of objects and intrusion detection [15–17]. Various advancements and methods have been proposed and implemented based on MAS to increase efficiency in surveillance. Milella et al. [6] implemented a MARS that is based on fixed and mobile agents for the active surveillance of places such as museums, airports, warehouses, etc. The system was able to detect the intrusion of persons within forbidden areas and send the location to the mobile agent, i.e., a robot, for further exploration of that area. Furthermore, Pennisi et al. [7] proposed an MRS for surveillance through a network of distributed sensors to detect a person through fixed sensors and send a robot to the location of the detected person for inspection and stopping the target person in case of any anomaly by blocking their way. Du et al. [18] presented a strategy for MAS-based surveillance to track an evader through cooperation between mobile agents. In another work, Mostafa et al. [19] proposed an autonomy model based on fuzzy logic to manage the autonomy of a MAS in complex environments. The aim of this model was to assist the autonomy management of the agents by helping them in making competent autonomous decisions. The application of this model was presented in the monitoring of movements of elderly people. In another work, Kariotoglou et al. [20] developed a framework based on stochastic reachability and hierarchical task allocation to solve the dimensionality problem faced by state-space gridding solutions based on dynamic programming for Markov decision processes in autonomous surveillance with a collection of pan-tilt cameras. The authors conducted the experiment with the proposed framework on a setup targeting industrial pan-tilt cameras and mobile robots. A MAS that includes robot as mobile agents is referred to as a MARS. During persistent surveillance through MARS, robots sequentially visit regions of interest (ROIs) based on applied constraints known as temporal logic (TL). Aksaray et al. [21] presented a method to minimize the time between visits of robots to ROIs by sharing the times of visits among them while considering their TLs to enhance efficiency and reduce redundant visits to those regions. Wu et al. [22] proposed an optimal method to sense robots based on less energy consumption for efficiently adjusting the position of mobile relay for maintaining the quality of the wireless link while the robots are moving. In another work, Jahn et al. [23] proposed a distributed technique for a team of robots to plan deformation while they are moving around a region to create a fence for perimeter surveillance and need to take this fence to another region. Scherer et al. [24] introduced multiple heuristics with various planning perspectives for convex-grid graphs and combined them with the

tree traversal approach for better communication in the MRS for persistent surveillance with connectivity constraints.

### 2.2. Role of Robotics during COVID-19

The outbreak of the COVID-19 pandemic has negatively impacted our society. This situation is inevitable and requires modern solutions. COVID-19 has interrupted our usual face-to-face interactions and frustrated us because of the possibility of spreading the virus through physical interaction. The fourth industrial revolution, known as Industry 4.0, should fulfil the requirements to effectively control and manage the COVID-19 pandemic [25]. The presence of intelligent robots surged in various fields, e.g., autonomous driving, medical, rehabilitation, education, companionship, surveillance, information guide, telepresence, etc., to minimize the potential spread of this virus [26]. Robot-assisted surgeries are also being taken into positive consideration in surgical environments [8,27]. Furthermore, Mahdi Tavakoli et al. [28] presented an analysis of the robotics and autonomous systems for healthcare during COVID-19. Based on their analysis, they recommended immediate investment in robotics technology as a good step toward making healthcare services safe for both patients and healthcare workers. Moreover, the ongoing pandemic is affecting the social well-being of people and triggering feelings of loneliness in them. Social and companion robots have been considered as a potential solution to mitigate these feelings of loneliness through continuous social interaction with less fear of spreading the infectious disease [29–33]. Rovenso recently developed a UV disinfectant robot targeting offices and commercial spaces [34]. Moreover, there is another autonomous robot named AIMBOT developed by UBTECH Robotics that performs disinfection tasks at Shenzhen Third Hospital [35].

### 2.3. Effectiveness of Physical Distancing

Multiple works have simulated the spread of the virus [36–38] to show the effectiveness of different social distancing measures. The ratio between the total cases of infections during the entire course of the outbreak is termed as the attack rate [39]. According to Mao [36], the attack rate can be decreased up to 82% if three consecutive days are eliminated from working days of a workplace setting. Within the same setting, the attack rate can be reduced up to 39.22% [37] or 11–20% by maintaining a physical distance of 6 feet between the individuals at the workplace depending upon the frequency of contact between them [38].

### 2.4. Emerging Technologies to Monitor Physical Distancing

Recently, different methods have been proposed to monitor the physical distancing between people. Workers in the warehouses of Amazon are monitored through CCTV cameras to detect physical distancing breaches [40]. Other techniques are based on the use of wearable devices [41,42]. These devices use the technologies of Bluetooth or ultra-wide band (UWB). Moreover, different companies such as Google and Apple are developing applications to trace the contacts of people so that alert messages can be delivered to the users if they come in close contact with an infected person [43]. In an extensive survey by Nguyen et al. [39], the technologies that can be used to track people to detect if they are following social distancing rules properly are discussed. The pros and cons of these technologies, such as WiFi, RFID, Bluetooth, artificial intelligence and computer vision, are also discussed in this comprehensive survey.

## 3. Proposed System

In this section, we first describe the hardware architecture used to build the proposed system. Then, we present our method with a detailed description about each functional module.

### 3.1. Hardware Architecture

We developed the robot on top of the smart base named 'EAIBOT SMART' [44], which is shown in Figure 2. The smart base consists of dual liDAR sensors of type YDLIDAR G4 to map the surrounding area. One of the liDAR sensors was mounted on top while the second was mounted beneath the smart base. Dual liDAR sensors mounted at two different heights made the collision avoidance and mapping system more robust. Collision avoidance was aided by a gyroscope and five ultrasonic sensors mounted in different directions to cover the world at 360°. We used built-in collision avoidance, mapping and navigation in this robot base. Moreover, the smart base had a 10-h battery life, which was long enough for it to survive for longer durations. It had a docking station as well and was able to autonomously navigate to the docking station to automatically put itself on charge.

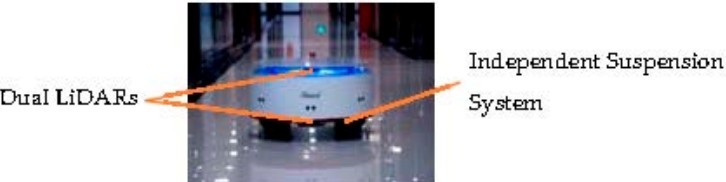

**Figure 2.** EAIBOT SMART.

We equipped our robot with a laptop on top of it to use its RGB camera, speaker and microphone for HRI. This laptop can be replaced with a tablet or any other setup having the sensors mentioned above for HRI. Furthermore, we set up the CCTV RGB cameras with the resolution of 1080, mounted at heights that provide angled views of different locations of the building so as to monitor different areas. We used a machine with the Intel i7 10th generation CPU and an Nvidia RTX 2070 Super GPU to process video streams received from the CCTV cameras.

### 3.2. Our Method

We used the Robot Operating System (ROS) [45] with the 'kinetic' distribution named to build our system. The ROS is very efficient in structuring and managing robot applications. Moreover, it ensures a modular and expandable system. The overall system architecture is shown in Figure 3.

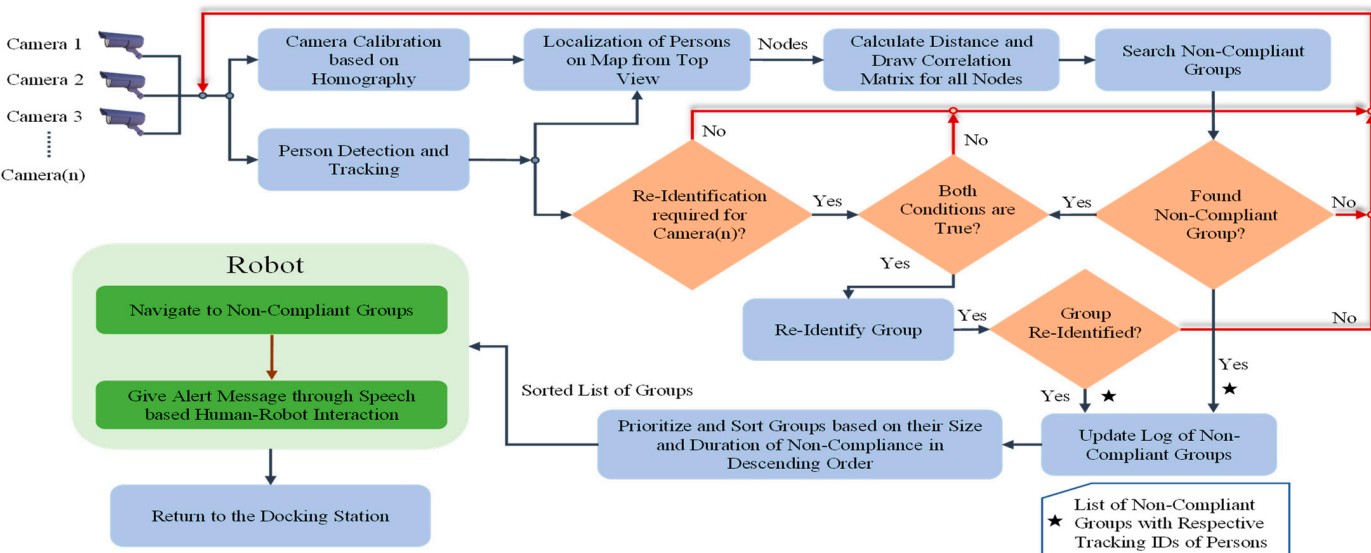

**Figure 3.** Overall system architecture.

In this section, we describe the main components of our method to detect the violation of physical distancing constraints and enforce them using HRI. The main components of our method are as follows: (1) Detection, localization and tracking of the persons; (2) Search for non-compliant groups based on the violation of physical distancing constraints; (3) Re-identification of the moving non-compliant groups through a multi-camera system; (4) Prioritization of the non-compliant groups; (5) Delivery of alert message to non-compliant groups through speech-based HRI.

### 3.2.1. Monitoring Physical Distancing Constraints

The criterion used for detecting the violation of physical distancing constraints by non-compliant groups in our method was to detect the physical distance of less than 1 m. All CCTV cameras continuously monitored the environment within their respective fields of view (FoV) to detect non-compliant groups. The main components of this functional module are as follows.

### Person Detection and Tracking

Object detection [46,47], localization and tracking have been active areas of research. For person detection and tracking, we used a pipeline based on the tracking algorithm proposed in [48] and the object detection method named 'You Only Look Once (YOLO)' version four, i.e., YOLOv4 [49]. According to the results presented in [49], it outperformed state-of-the-art object detection methods such as 'EfficientDet' [46] and its own previous version named 'YOLOv3' [47] in terms of the average precision and frames per second (FPS) speed. The experimental results showed that the pipeline achieved very good performance in terms of accuracy and speed. Input to this pipeline was RGB images received from the CCTV camera and output was the set of bounding boxes, i.e., top left corner coordinates, width, and height, for the detected persons in the given image. It also generated a unique identity for each detected person that remains same while the person remains in the current FoV of the camera. In order to consume this pipeline for video streams from each CCTV camera within the multi-camera system for monitoring, we used multi-threading and asynchronous calls. Each video stream was handled in an independent thread.

### Localization of Detected Persons

All CCTV cameras were mounted in such a way that they provided angled views of the ground plane. In order to accurately calculate the distance between the persons, we preferred the top view of the ground plane, which represents the exact location of persons' feet on ground. For this purpose, we converted the angled view to the top view by applying the homography matrix to the four reference points on the angled view from CCTV. These reference points were manually selected during camera calibration so that they could cover the maximum area of the FoV of the CCTV camera, as shown in Figure 4.

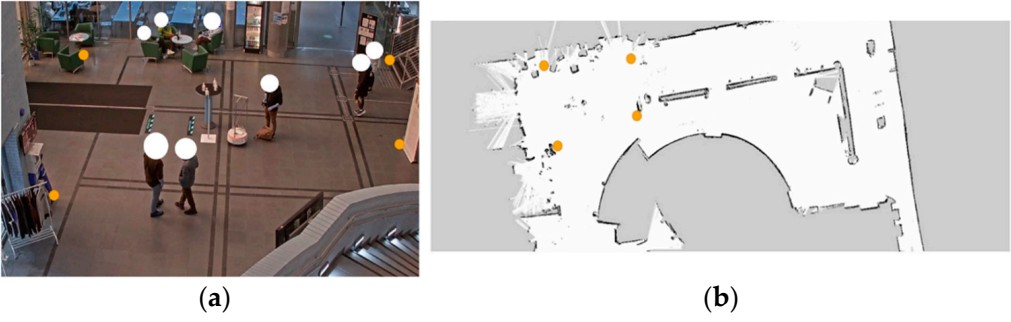

|  (a)  |  (b)  |

**Figure 4.** CCTV camera calibration: Yellow markers represent (**a**) pixel coordinates in angled view; (**b**) Cartesian coordinates in top view.

The conversions of these points were performed by using Equation (1).

$$\begin{bmatrix} x_{topView} \\ y_{topView} \end{bmatrix} = H * \begin{bmatrix} x_{angledView} \\ y_{angledView} \end{bmatrix} \tag{1}$$

In Equation (1), $x_{angledView}$ and $y_{angledView}$ indicate the pixel coordinates of one of the four reference points in the angled image view from the CCTV; $x_{topView}$ and $y_{topView}$ represent the same point after conversion to top view; and '$H$' represents the scaled $3 \times 3$ homography translation-matrix. To transform the angled view to the top view of any detected person, as shown in Equation (2), we used the middle point of the bottom corners' points of the bounding box yielded by the detection and tracking pipeline (Section 'Person Detection and Tracking') against that detected person. This middle point represented the feet of the detected person.

$$\left[ x_{topView}^{P(k)}, y_{topView}^{P(k)} \right] = H * \left[ x_{angledView}^{P(k)}, x_{angledView}^{P(k)} \right] \tag{2}$$

where $\left[ x_{angledView}^{P(k)}, y_{angledView}^{P(k)} \right]$ represents the pixel coordinates of the feet of the detected person $P(k)$ with a unique ID '$k$' in the angled image view from the CCTV, and $\left[ x_{topView}^{P(k)}, y_{topView}^{P(k)} \right]$ represents the same point after conversion into top view.

Distance Estimation and Search for Non-Compliant Groups

After transforming the pixel coordinates of the detected persons into top view, we estimated the distance between each person. Here, we treated each transformed position as a node. The distance between two nodes was calculated using the formula of Euclidean distance. The pair-wise Euclidean distances between multiple persons are shown in Table 2.

**Table 2.** Euclidean distances between persons.

| $P_k$ | $P_1$ | $P_2$ | $P_3$ |
|-------|-------|-------|-------|
| $P_1$ | 0 | 1 | 2 |
| $P_2$ | 1 | 0 | 1 |
| $P_3$ | 2 | 1 | 0 |

Then, a truth table was created based on this correlation matrix of Euclidean distances. Any connection or Euclidean distance less than one meter between the two nodes was denoted as *True* (Table 3).

**Table 3.** Truth table based on the correlation matrix.

| $P_k$ | $P_1$ | $P_2$ | $P_3$ |
|-------|-------|-------|-------|
| $P_1$ | 1 | 1 | 0 |
| $P_2$ | 1 | 1 | 1 |
| $P_3$ | 0 | 1 | 1 |

A modified depth-first search algorithm [50,51] was used to find all paths between the nodes in the environment, with no repeated nodes. In an environment with only one path, the algorithm can find this path in '$O(V + E)$' time, where '$V$' and '$E$' represent the number of vertices and edges in the graph, respectively. However, there is a possibility of a very large number of paths in a graph, for instance, '$O(!n)$' in a graph of order '$n$'. In order to deal with this problem, the search on each path was terminated when it reached the threshold of group size '$x_c$', where '$c$' represents the number of nodes, i.e., persons, in the group. In our case, we considered '$x_c = 10$' as a threshold value to stop searching. For each returned path, the average position of all the nodes in a group was considered as the

position of that group in the map. This information about the position of a non-compliant group was used to navigate the robot. Representation of the list of non-compliant groups of detected persons is shown in Equation (3).

$$G = \{G_l \mid G = [\![l, P_k, \ldots \ldots, P_c, t_l]\!]\} \tag{3}$$

where '$l$' is the unique identity assigned to the group '$G_l$', '$k$' is the unique identity of the person '$P_k$', '$c$' is the total number of nodes and '$t_l$' is the duration of non-compliance by the $l$th non-compliant group '$G_l$'. '$t_l$' was computed by the tracking of non-compliant groups based on unique identities given to the tracked persons.

### 3.2.2. Re-Identification of Non-Compliant Groups

This module of the system ensured the persistent monitoring and tracking of the non-compliant groups that were moving within the environment through a multi-camera setup. It was performed using the motion tracking and the re-identification modules based on the information communication between multiple CCTV cameras through the server. Re-identification was performed in parallel to the detection, tracking and localization of the non-compliant groups, as shown in Figure 3. The duration of non-compliance ($t_l$) by a moving non-compliant group kept incrementing until that group was re-identified while again violating the physical distancing constraints. The flow diagram in Figure 5 shows the overall process of group re-identification.

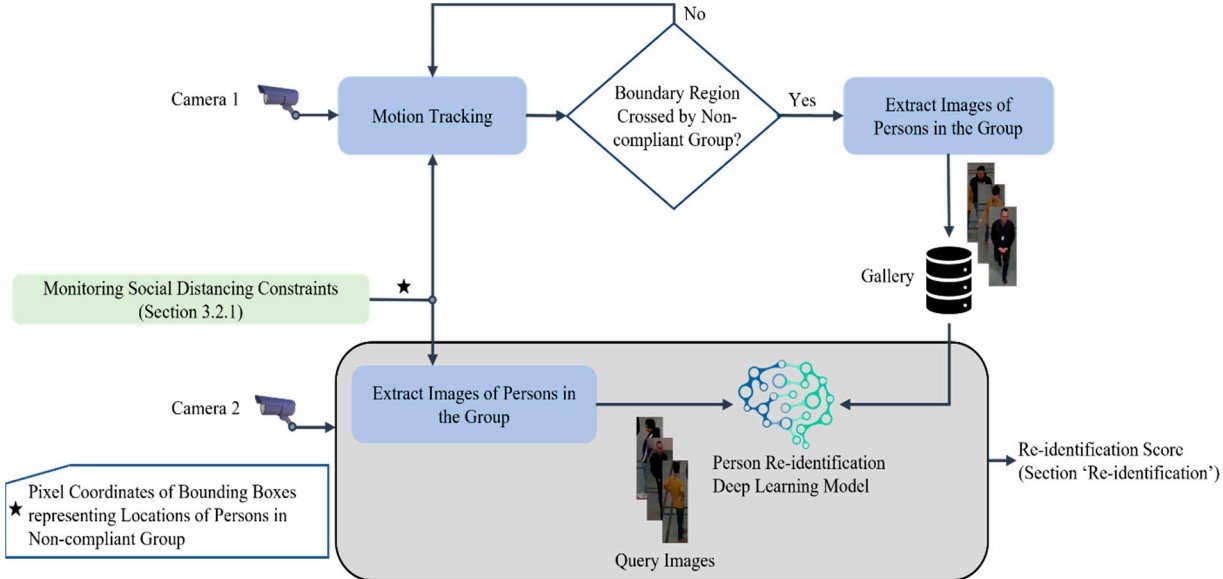

**Figure 5.** Re-identification of non-compliant groups.

Motion Tracking

In the process of the motion tracking of each non-compliant group, we first selected bounding boxes based on the identities of the detected persons belonging to a non-compliant group, yielded by the section 'Distance Estimation and Search for Non-Compliant Groups'. After selecting the bounding boxes, we found the center position of the non-compliant group in an image received from the CCTV camera by taking the average of the pixel coordinates representing the top leftmost and bottom rightmost corners among all corners of the bounding boxes selected in the previous step. We repeated the previous two steps with the skip of five frames in the video stream to find the position of that group in the next frame. After finding the positions of the group in two different frames, we calculated the Euclidean distance (Equation (3)) between pixel coordinates at both positions of the group. It was considered as a change in position if the calculated distance reached a

set threshold value. We divided the FoV of the CCTV camera into boundary regions, which is shown in Figure 6.

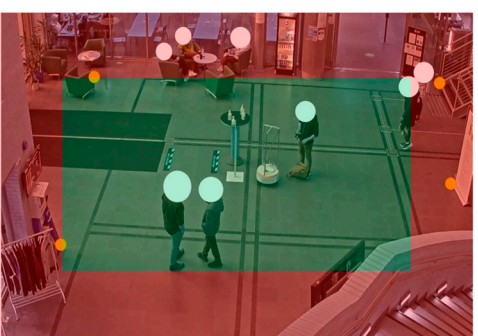

**Figure 6.** Division of the FoV of a CCTV camera into regions.

If the new position of the moving non-compliant group was located within the boundary regions, images of the persons of that group were extracted based on the bounding boxes yielded by the person detection pipeline, which is discussed in the section 'Person Detection and Tracking'. Later, these images were stored in the database to re-identify that non-compliant group in the case of being detected through other CCTV cameras while again violating the physical distancing constraints. Information kept in the name of each stored image of a person was person identity and group identity as '$P_{lk}$'.

Re-Identification

We developed a pipeline for group re-identification based on the existing algorithm for person re-identification. In order to perform the re-identification of non-compliant groups, we used a lightweight and state-of-the-art person re-identification deep learning model termed as Omni-Scale Network (OSNet) [52]. We used a pre-trained model, which was trained on six widely used person re-identification image datasets: Market1501 [53], CUHK03 [54], MSMT17 [55], DukeMTMC-reID (Duke) [56,57], GRID [58] and VIPeR [59]. In this person re-identification technique, the database of existing images of the persons, from which the model has to re-identify the target person, is called 'Gallery', and the image of the target person is called the 'Query' image. For implementation, we used a library for deep learning person re-identification named Torchreid [60], which is based on a well-known machine learning library named PyTorch.

Group re-identification was performed whenever a non-compliant group was detected through any CCTV camera to re-identify if it was previously detected by another camera while violating the physical distancing constraints. We cropped and extracted images of the persons in non-compliant groups detected by the person detection and tracking pipeline (Section 'Person Detection and Tracking') through a current CCTV camera for using them as query images in the re-identification module. The Gallery was based on the images of the persons present in moving non-compliant groups, which were tracked and extracted by the motion tracking module (Section 'Motion Tracking') We used the term re-identification score 'Sl' to determine if the target non-compliant group was re-identified, where 'l' represents the unique identity of that group. It was based on how many persons were re-identified from the group based on the query images. '$S_l = 1$' meant that one person from the query images was re-identified. The target non-compliant group was considered as re-identified in case of '$S_l \geq 2$', which means that at least two persons were re-identified from query images. The steps included in the pipeline developed for performing group re-identification are shown in Algorithm 1.

---

**Algorithm 1** Re-identification of non-compliant group

---

**Input:** $[P_{lk}, \ldots\ldots, P_{ln}]$ = Extracted images of persons in target non-compliant group as
**Output:** Re-identification Status (*True* or *False*), $[IDP_{lk}, \ldots\ldots, IDP_{ln}]$ = List of identities of re-identified persons as $IDP_i$
**Steps:**
**1:** $OSNet \leftarrow P_i, D$

**2:** $[P_{lk}, \ldots\ldots, P_{ln}] \leftarrow OSNet$
**3:** Compute $S_l$
**4:** If $S_l \geq 2$ then return *True*, $IDP_i$
**5:** Else return *False*, $IDP_i$

---

Where '$l$' represents the identity of the group, '$k$' represents the identity of the person and '$D$' represents the Gallery database of the persons belonging to the moving non-compliant groups. Whenever a non-compliant group was re-identified, the corresponding identities of the persons from that group were updated in the database to keep track of them.

### 3.2.3. Prioritization of Non-Compliant Groups

Prioritization refers to which non-compliant group should be addressed first by the robot. Only those non-compliant groups were considered in this step who were locked at a location and not moving, which was decided based on the section 'Motion Tracking'. It was performed based on two factors, namely, the size of the non-compliant group ($x_c$) and the duration of non-compliance ($t_l$). Prioritization was performed in a hierarchical way based on these two factors. '$t_l$' was considered first while prioritizing the non-compliant groups because the continuous violation of physical distancing constraints over a longer period can be more dangerous. '$t_l$' was divided into different ranges with a constant difference. It could be '0 min $\leq t_l \leq$ 5 min', '5 min $\leq t_l \leq$ 10 min', etc. The non-compliant groups within the higher range of '$t_l$' were given high prioritization based on '$x_c$'. The group with a higher value of '$x_c$' was given higher priority. On top of these two factors, a non-compliant group that was re-identified while again violating physical distancing constraints was given highest priority due to its continuity in violation.

### 3.2.4. Enforcement of Physical Distancing Constraints

After the prioritization of locked non-compliant groups, a prioritized list of locations of these groups was transmitted to the social robot for sending it to these locations one by one and giving alert messages to these non-compliant groups through speech-based HRI. The robot received an updated list of locations of non-compliant groups after a fixed interval of time (i.e., 5 s) after every search for non-compliant groups. Once the robot arrived in the vicinity of the location of a group, it played an audio message to alert the persons in the group about violation and encourage them to maintain a physical distance of at least one meter. Moreover, the robot explained to them the reason that they were approached by it was because they were not abiding by the rule of maintaining a safe physical distance.

## 4. Experimental Results

In this section, we explain the experimental setup and the metrics used to evaluate the proposed system and provide an analysis of the results obtained during experiments. We tested our system in the main building of our university (Norwegian University of Science and Technology (NTNU)) with a total of 15 participants divided into four different groups. All groups were given demonstrations of the test scenarios, which are discussed in Sections 4.1 and 4.2 Three cameras were mounted in different locations of the building to monitor three different areas. Overall, 28 experiments were conducted to test the proposed system. The number of violations detected and the successful enforcements performed under different configurations are shown in Table 4.

**Table 4.** Overall performance of proposed system with respect to different configurations.

| Configuration | Total Experiments (Violations Made) | Number of Violations Detected | | Number of Enforcements |
|---|---|---|---|---|
| | | Single Group at a Time | Multiple Groups at a Time | |
| Without Group Re-identification | 19 | 11 | 6 | 17 |
| With Group Re-identification | 9 | 8 | | 8 |

In overall experiments, the system was unable to properly detect the violation on three occasions. In two of them, the reason was that one individual was fully occluded by the other person standing in front them in-line with the camera's angled view in a two-person group. The third system failure was due to the failure in group re-identification, which occurred due to a large variation in lighting conditions. However, the violation was detected, and enforcement was performed precisely. Details of these experiments and metrics used to evaluate the proposed system and their results are discussed as follows:

### 4.1. Accuracy of Non-Compliant Group Localization

This evaluation metric was used to measure the accuracy of localization based on comparison between the ground truth location of the non-compliant group and the location estimated using our method. Prior to the experiments, ground truth locations were manually marked on the 2D map of the ground plane based on the Cartesian coordinate system used by the robot for navigation and localization. Participants were planted at those ground truth locations in order to test the accuracy of our system with respect to the localization of non-compliant groups. The plot of the ground truths as green circles and estimated locations of non-compliant groups as orange circles is shown in Figure 7.

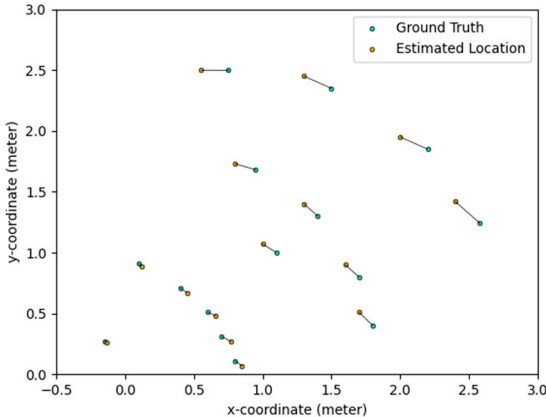

**Figure 7.** Localization of non-compliant groups.

The above plot shows the non-compliant groups being localized with respect to the Cartesian coordinate system fixed to the robot. The maximum error observed between the ground truth and the estimated location was 0.24 m and the average error observed between them was approximately equal to 0.12 m. The detection and localization of the non-compliant groups during two different experiments performed with the proposed system and navigation of the robot to these groups are shown in Figure 8.

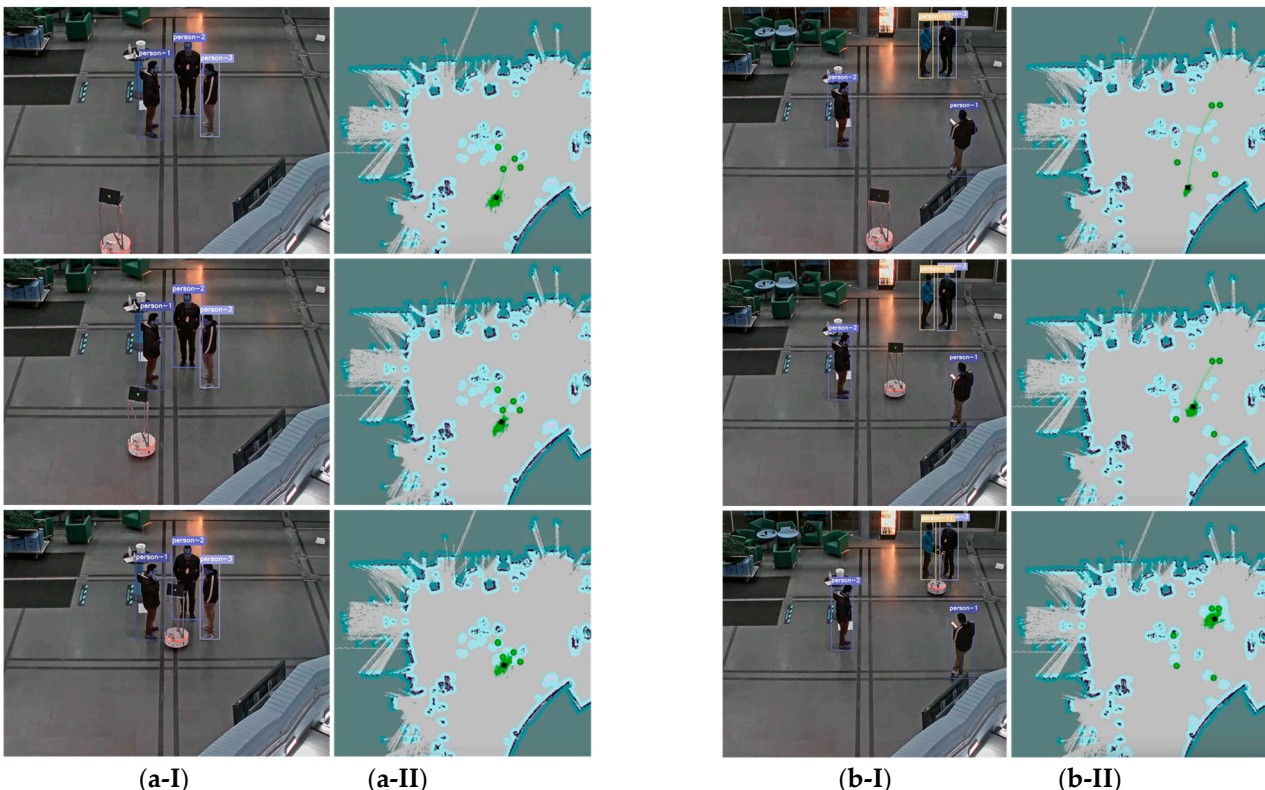

|           | (**a-I**) | (**a-II**) |           | (**b-I**) | (**b-II**) |

**Figure 8.** Localization of non-compliant groups and navigation of robot to give the alert message. (**a-I**,**a-II**) represent the experiment based on three persons, and (**b-I**,**b-II**) represent the experiment based on two persons in a non-compliant group. (**a-II**,**b-II**) show the navigation and localization of the robot and the persons belonging to non-compliant group on the map. Green and black markers on map represent the persons and the robot, respectively.

### 4.2. Accuracy of Re-Identification of Non-Compliant Groups

We designed multiple scenarios in order to test the accuracy of the proposed system in the re-identification of non-compliant groups and the enforcement of physical distancing constraints. As shown in Table 4, eight experiments were based on group re-identification. Groups of participants were asked to violate physical distancing constraints within the FoV of one of the three mounted CCTV cameras and then walk to the other side and perform the same action within the FoV of another CCTV camera. Here, the purpose was to re-identify the non-compliant groups while repeating the violation of physical distancing constraints. We stored the video streams captured by all the mounted CCTV cameras while conducting the experiments for creating our own dataset and testing the overall accuracy of the re-identification deep learning model with respect to the environmental conditions in our designed test scenarios. We used YOLOv4 [49] to crop the images of the persons from all video frames and then annotated each person with a personal as well as camera identity. In this way, we created our own dataset, which consisted of 7687 images of 15 different persons captured through three different CCTV cameras. After annotation of the dataset, we divided it into two categories: Gallery and Query images. Ten percent of the total images were categorized as Query images and rest were used as Gallery images. The results of the re-identification module based on the deep learning model on our created dataset are shown in Table 5.

**Table 5.** Accuracy of re-identification module on our dataset.

| R1 (%) | R5 (%) | R10 (%) | mAP (%) |
|--------|--------|---------|---------|
| 99.8   | 100    | 100     | 95.1    |

Figure 9 shows the enforcement of physical distancing constraints based on the re-identification of a non-compliant group, and Figure 10 presents some of the results that were generated while testing group re-identification.

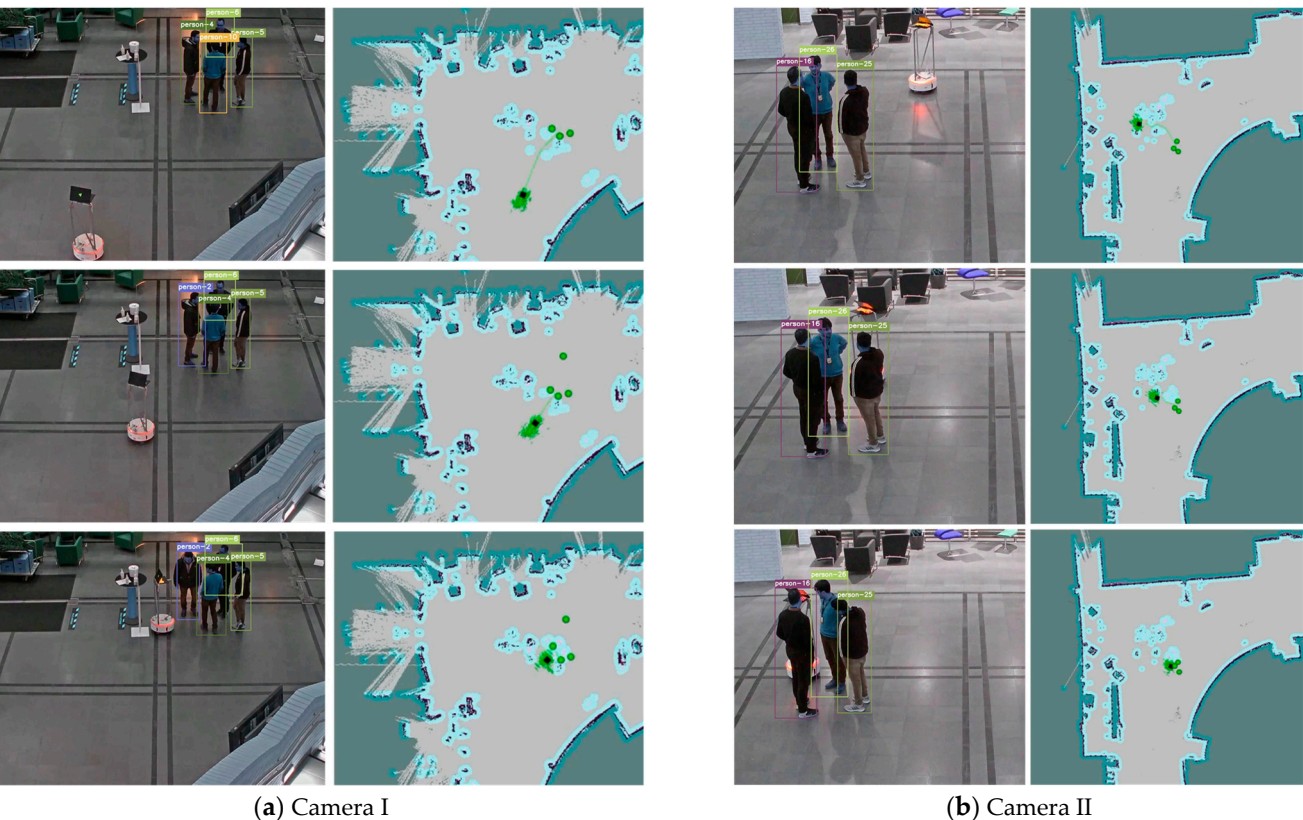

(**a**) Camera I　　　　　　　　　　　　　　　　(**b**) Camera II

**Figure 9.** Re-identification of non-compliant group and enforcement of physical distancing constraints through robot. (**a**) A non-compliant group consisted of four persons detected by Camera I. (**b**) Same group re-identified by Camera II with re-identification score, while again violating the physical distancing constraints.

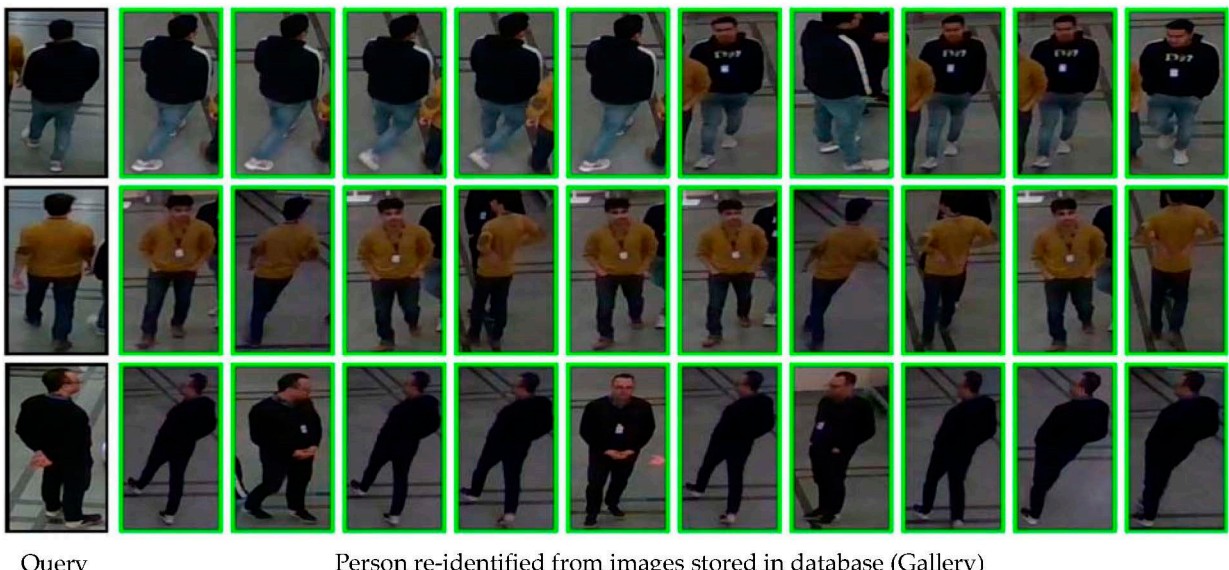

Query　　　　　　　Person re-identified from images stored in database (Gallery)
image

**Figure 10.** Images of persons re-identified by re-identification module based on images stored in database (Gallery) and given query images.

## 5. Conclusions, Limitations and Future Work

A novel system for monitoring and enforcing physical distancing constraints in large areas is proposed in this paper, which consists of multiple CCTV cameras and an MRS. Monitoring was conducted using a multi-camera system to detect and track groups of persons who did not comply with physical distancing constraints. We proposed a pipeline for group re-identification to detect repetitive violations of physical distancing constraints by a non-compliant group of individuals. We used an autonomous, collision-free mobile robot for the enforcement of physical distancing constraints by attending to non-compliant groups through HRI and encouraging them to comply with the set constraints. The effectiveness and accuracy of our system were demonstrated in terms of the detection and localization of non-compliant groups, group re-identification in the case of repetitive non-compliance and the enforcement of physical distancing constraints through HRI. We conclude that the monitoring of physical distancing constraints with group re-identification is effective in the long-term tracking of non-compliant groups to detect repetitive violations and notify the security control room in a timely manner to stop them. We also considered the ethical concerns in our system through efficient and secure data gathering and data handling mechanisms.

A limitation of our system is that the re-identification of the non-compliant group is not deployed by the robot. The re-identification of the non-compliant group through the robot would increase the overall accuracy of the system with regard to the enforcement of physical distancing constraints. Due to COVID-19 restrictions, we could only test the system in controlled settings with a low crowd density. Moreover, due to the same reason, we could not evaluate the social impact of our system.

In the future, testing our system in environments with high crowd densities is required to make it more robust. Furthermore, usability tests with security officials need to be conducted to demonstrate the effectiveness of the proposed system. In future studies, we will develop a mechanism based on our MARS to predict the future location from the past motion trajectory of a non-compliant group that is wandering within the environment. This can help to attend to non-compliant groups that are wandering within the area in a timely manner.

**Author Contributions:** Conceptualization, S.H.H.S. and I.A.H.; methodology, S.H.H.S., O.-M.H.S. and E.G.G.; software, S.H.H.S., O.-M.H.S. and E.G.G.; validation, S.H.H.S., O.-M.H.S. and E.G.G.; formal analysis, S.H.H.S. and I.A.H.; investigation, S.H.H.S., I.A.H., O.-M.H.S. and E.G.G.; writing—original draft preparation, S.H.H.S., O.-M.H.S. and E.G.G.; writing—review and editing, S.H.H.S. and I.A.H.; supervision, S.H.H.S. and I.A.H. All authors have read and agreed to the published version of the manuscript.

**Funding:** This research received no external funding.

**Acknowledgments:** The authors are grateful to the Norwegian University of Science and Technology (NTNU) for their support.

**Conflicts of Interest:** The authors declare no conflict of interest.

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
