# Peer review of "Multi-Agent Robot System to Monitor and Enforce Physical Distancing Constraints in Large Areas to Combat COVID-19 and Future Pandemics"

_applsci, doi:10.3390/app11167200_

Round 1

Reviewer 1 Report

The idea of an autonomous monitoring system that is able to enforce social distancing rules in large areas round the clock without human intervention is a nice one, and the system is clearly described and presented.

I have some concerns about some decisions made by the Authors, especially to detect rules braking on the group (and not - individual or dyads) level, as well as use tracking of a group, rather than individuals who braked the rule. The idea of calculation of the duration of rule braking (and the number of group members) rather than looking for full enforcement possibilities  (calculation minimal time to access the group by Rumbas and weight the seriousness of braking the rule by for example by the total minutes of rule braking in the dyads, multiply by numbers of dyads in the group to make the order in the line) can be discussable, but it is rather an example than a monitoring and enforcement instrument.

I also assume that Euclidean distance between persons are calculate based on minimal distance between any part of body, which could be misleading for the purpose of the regulation (where distance between heads is important). I would include the levels in the distance braking a part of weight of the group (or dyad) importance. But these are only additional thinking for the Authors, rather than objections.  

I would like to have more data in the tab. 4: how many trial were done, how many were false and for what reasons. There is a need to check how the system will be working in not so easy conditions – many groups, groups that change the members or moves rapidly and the individual distances between them, because for now I have the feeling of the Authors being happy with group re-identification and individual identification (which is not very exiting). So, I would recommend the cross the "pandemia limitation" and work more on testing the system.

From the other hand side there to many details and space in the article for obvious calculations (3.2.1.3).

Author Response

Comment: 1

“I have some concerns about some decisions made by the Authors, especially to detect rules braking on the group (and not - individual or dyads) level, as well as use tracking of a group, rather than individuals who braked the rule. The idea of calculation of the duration of rule braking (and the number of group members) rather than looking for full enforcement possibilities (calculation minimal time to access the group by Rumbas and weight the seriousness of braking the rule by for example by the total minutes of rule braking in the dyads, multiply by numbers of dyads in the group to make the order in the line) can be discussable, but it is rather an example than a monitoring and enforcement instrument.”

Respected reviewer suggests the idea of calculation of the duration of rule braking (and the number of group members) for prioritization of non-compliant groups. As discussed in Section 3.2.3. of our manuscript, we prioritized the non-compliant groups based on two factors: 1) the size of groups, where size refers to the number of persons present in a particular non-compliant group and 2) duration of non-compliance by a group of individuals. We believe that it satisfies the reviewer’s concern in above mentioned comment.

Comment: 2

“I also assume that Euclidean distance between persons are calculate based on minimal distance between any part of body, which could be misleading for the purpose of the regulation (where distance between heads is important). I would include the levels in the distance braking a part of weight of the group (or dyad) importance. But these are only additional thinking for the Authors, rather than objections.”

As discussed in Section 3.2.1.2. Line 240 of our manuscript, the distance between the persons was calculated based on the distance between feet of two persons which represent their exact location on the ground. In order to make it more clear, we have made changes at Line 229 to explicitly mention it.

Comment: 3

“I would like to have more data in the tab. 4: how many trial were done, how many were false and for what reasons. There is a need to check how the system will be working in not so easy conditions – many groups, groups that change the members or moves rapidly and the individual distances between them, because for now I have the feeling of the Authors being happy with group re-identification and individual identification (which is not very exiting). So, I would recommend the cross the "pandemia limitation" and work more on testing the system.”

As suggested by reviewer, we have added complete details of experiments in Table 4. Moreover, we have explained the reasons for system’s failures right after Table 4 at Line 347. Our proposed system also performs in the conditions of multiple non-compliant groups as mentioned in Section 3.2.1.3. Equation 3 and Section 3.2.4. We tested our system with multiple non-compliant groups at a time and now explicitly mentioned the experimental details in Table 4. We mentioned in our limitations that it was difficult to test the system in places with high crowd density due to COVID restrictions and lockdown. However, we tested the system with most possible scenarios related to breach in physical distancing rules. The search for non-compliant groups was made after every 5 seconds and updated list with information about the non-compliant groups was sent to the robot which hopefully satisfies the reviewer’s concern about change or movement in the group members. We added the detail in Section 3.2.4. Line 331.

Comment: 4

“From the other hand side there to many details and space in the article for obvious calculations (3.2.1.3).”

We removed the unnecessary details i.e., Equation 3 from the Section 3.2.1.3.

Reviewer 2 Report

The manuscript „Multi-Agent Robot System to Monitor and Enforce Social Distancing Constraints in Large Areas to Combat COVID-19 and Future Pandemics“ is approximately 9100 words long. The manuscript is divided into five sections, the first being the introduction. Then follows the overview of existing work applying MAS to tackle epidemics. The third section describes the proposed surveillance and agent system. In section four, evaluation metrics and experimental results are described, and in the final section, the authors discuss the system’s limitations and future research directions.

The proposed system uses multi-agent systems for surveillance and physical distance keeping as a means to counteract COVID-19 spreading. A series of 25 experiments are conducted to test MAS efficiency in detecting and localizing non-compliant groups, warning them, and then detecting if repetitious distance violations happened. Results suggest a successful technology application.

Previous research and relevant references are clearly stated. The methodology is articulate, as well as the results. Therefore, I suggest publishing the manuscript. No plagiarism is detected using dedicated software.

Some minor corrections are suggested, though:

[line 41] there is redundancy in mentioning Table 1 in this lines 39 and 41; expunge the end of the sentence „which is summarized in Table 1.“

[48] „social distancing“ is a concept in sociology referring to a minority group’s integration level into a community or society; „physical distancing“ is a more appropriate phrase

[152] “COVID-19 has almost destroyed face-to-face human contact…” is clearly an overstatement. It certainly interrupted our usual face-to-face interactions and frustrated us both due to its intensity and duration.

Additionally, in the final section of the paper, a sentence should be added stating that ethical concerns for applying such technology are also considered.

Author Response

Comment: 1

“[line 41] there is redundancy in mentioning Table 1 in this lines 39 and 41; expunge the end of the sentence which is summarized in Table 1.”

We have removed the redundant information.

Comment: 2

“[48] ‘social distancing’ is a concept in sociology referring to a minority group’s integration level into a community or society; „physical distancing“ is a more appropriate phrase”

As suggested, we have replaced the phrase ‘social distancing’ with ‘physical distancing’ throughout the manuscript including title.

Comment: 3

“[152] “COVID-19 has almost destroyed face-to-face human contact…” is clearly an overstatement. It certainly interrupted our usual face-to-face interactions and frustrated us both due to its intensity and duration.”

We have revised the mentioned line and changed it into the line as follows: “COVID-19 has interrupted our usual face-to-face interactions and frustrated us because of the possibility of spreading the virus through physical interaction.”

Comment: 4

Additionally, in the final section of the paper, a sentence should be added stating that ethical concerns for applying such technology are also considered.

As recommended, we have added the sentence related to the ethical concerns at Line 395 in Section 5. The added line is as follows: “We also considered the ethical concerns in our system through efficient and secure data gathering and data handling mechanisms.”
